# Hierarchical Optimization of 3D Point Cloud Registration

**DOI:** 10.3390/s20236999

**Published:** 2020-12-07

**Authors:** Huikai Liu, Yue Zhang, Linjian Lei, Hui Xie, Yan Li, Shengli Sun

**Affiliations:** 1Shanghai Institute of Technical Physics, Chinese Academy of Sciences, Shanghai 200083, China; liuhuikai@mail.sitp.ac.cn (H.L.); zhangyue8@mail.sitp.ac.cn (Y.Z.); leilj@shanghaitech.edu.cn (L.L.); xiehui@mail.sitp.ac.cn (H.X.); liyan@mail.sitp.ac.cn (Y.L.); 2School of Electronic, Electrical and Communication Engineering, University of Chinese Academy of Sciences, Beijing 100049, China; 3School of Information Science and Technology, ShanghaiTech University, Shanghai 201210, China

**Keywords:** 3D point cloud registration, improved voxel filter, multi-scale voxelized GICP

## Abstract

Rigid registration of 3D point clouds is the key technology in robotics and computer vision. Most commonly, the iterative closest point (ICP) and its variants are employed for this task. These methods assume that the closest point is the corresponding point and lead to sensitivity to the outlier and initial pose, while they have poor computational efficiency due to the closest point computation. Most implementations of the ICP algorithm attempt to deal with this issue by modifying correspondence or adding coarse registration. However, this leads to sacrificing the accuracy rate or adding the algorithm complexity. This paper proposes a hierarchical optimization approach that includes improved voxel filter and Multi-Scale Voxelized Generalized-ICP (MVGICP) for 3D point cloud registration. By combining traditional voxel sampling with point density, the outlier filtering and downsample are successfully realized. Through multi-scale iteration and avoiding closest point computation, MVGICP solves the local minimum problem and optimizes the operation efficiency. The experimental results demonstrate that the proposed algorithm is superior to the current algorithms in terms of outlier filtering and registration performance.

## 1. Introduction

Three-dimensional point cloud registration is a fundamental task for many 3D computer vision applications, such as 3D reconstruction [1], 3D recognition [2], and simultaneous localization and mapping (SLAM) [3]. The purpose of registration is to transform a set of point clouds in various views into the same coordinate system, which is optimal for model recovery or pose estimation.

ICP [4] and its variants are the most widely used 3D point cloud registration methods due to their simplicity and good performance. However, considering the closest point as the corresponding point may put the ICP method at risk of the local optimum issue, especially when the point cloud has outliers or does not have good initialization. In addition, ICP cannot process a large scale of input clouds because of its high computational complexity.

With the advancement of sensor technology, it has become easier and cheaper to obtain 3D point clouds, making point clouds more widespread. However, the scanned point clouds often have a huge scale and outliers, so they need to be processed before registration. The voxel filtering [5] can quickly and uniformly reduce the scale of the point cloud, while it is not good at removing outliers. Methods based on statistics [6,7], radius [8] and point density [9] remove random outliers by calculating the distribution of neighboring points and setting appropriate thresholds. However, they cannot work well when many outliers are evenly distributed. In addition, they spend more time due to the neighborhood point calculation.

To solve the local minimum problem of ICP, some algorithms have been proposed. Go-ICP is the first global registration algorithm based on the ICP framework; it uses a branch and bound method to find the global optimal. Although it solves the local minimum problem, it is still sensitive to the initialization. A coarse-to-fine strategy [10,11] is presented, where the coarse registration provides an initial estimation for the fine registration to obtain a precise alignment. This strategy can accurately register point clouds without initialization. However, it consumes more time for two parts of the calculation, especially for the point cloud feature extraction in coarse registration.

In this paper, a hierarchical optimized 3D point cloud registration algorithm includes improved voxel filter and Multi-Scale Voxelized GICP (MVGICP) is proposed. We introduce an improved voxel outlier filter, which combines voxel downsampling and point density information, delete grids of which the number of points is less than the threshold, and replace other points with centroids. Compared with the traditional voxel filter and the statistical outlier removal method, the improved voxel filter has the best filtering performance and takes less time than the statistical algorithm. Then we propose Multi-Scale Voxelized GICP (MVGICP) to register the 3D point clouds, which is based on the distribution-to-distribution strategy. MVGICP firstly voxelizes the target point cloud with a certain voxel size and calculates the point cloud distribution in the grid. The point cloud distribution is then brought into the GICP framework to get the transformation, followed by repeating this process with a smaller voxel size until the size is small enough. Thus, MVGICP does not require any coarse registration. Larger voxel size can initially transform the point cloud and the small size can further refine the registration result. MVGICP does not need the closest neighbor search either, so it can significantly improve the computing efficiency. The experimental results show that the proposed algorithm can effectively filter out the outliers and obtain better registration accuracy as well as computation efficiency in several datasets. In general, the contribution of this paper can be summarized as follows:

(1) An improved voxel outlier filtering method is proposed. This method combines traditional voxel filters and point density to achieve accurate filtering of outliers while efficiently downsampling.

(2) A novel 3D point cloud registration algorithm MVGICP is presented. It uses multi-scale iteration to avoid the complex closest points computation, and significantly improves the situation that GICP is prone to fall into local minimums and low calculation efficiency. At the same time, it has higher registration accuracy than VGICP.

(3) A thorough evaluation of the proposed algorithm is presented, including comparisons with the current algorithms in terms of outlier filtering and registration performance. The experimental results show that the proposed method performs better than other algorithms.

The remainder of this paper is organized as follows. Section 2 reviews the related work of other researchers. Section 3 demonstrates the details of our proposed method. Section 4 provides the experiment results and analyses. Section 5 concludes with the summary and the perspectives.

## 2. Related Work

### 2.1. Point Cloud Filtering

Point cloud filtering is a process that takes place before point cloud registration. Since the dense 3D point cloud acquired by terrestrial laser scanners or RGB-D instruments is enormous and contains many outliers, it is necessary to remove outliers and downsample the raw data. Voxel downsampling [5] is the most common method. By dividing the point cloud by grid and replacing other points with the centroid, the voxel method achieves the best efficiency. However, it is not good at removing outliers. Statistical outlier removal [6] in the Point Cloud Library calculates the mean and standard deviation of the distance from each point to its neighborhood and then removes the points for which the distance is outside the set range. Yang et al. [7] added dynamic standard deviation thresholds to the statistical algorithm to solve the irregular density distribution. Pirotti et al. [9] proposed an improved statistical algorithm and local outlier factor algorithm based on K-nearest neighbors (KNN). The local outlier factor relies on the local density with respect to its neighbors.

### 2.2. Coarse Registration

When the point cloud sets start in an arbitrary initial pose, the local registration algorithms easily fall into a local minimum. Then, registration returns to solving a global problem. The coarse alignment can compute an initial estimate of the rigid motion between two surfaces. The manual method is firstly applied; Zhang et al. [12] proposed manually adding labels on the model before registration and then aligning them through the label feature information. However, this is not suitable for the complex point cloud. Principle component analysis (PCA) [13] is more reliable when the shape of the target point cloud and the source point cloud is the same. Huang et al. [14] integrated the projection strategy with the Fourier signal matching and promoted the performance of noisy and low overlap point clouds. Some studies introduced methods based on the RANSAC iteration. Aiger et al. [15] exploited the invariant property of a four-point congruent set. Theiler et al. [16] addressed a keypoint-based four-point congruent set (K-4PCS) to overcome the low efficiency of 4PCS. In addition, Mellado N et al. [17] greatly enhanced the 4PCS algorithm by smart indexing. Voxel-based 4PCS [18] voxelizes the point cloud and generates plane patches before extracting four-plane congruent sets. V4PCS improves the robustness to unequal point density or point clouds from different sources. Other studies based on the local geometric features of the point cloud are more extensive. Frome et al. [19] proposed 3D shape context (3DSC), which is an extension of 2D shape context. It adopts a feature description method based on shape contour, which uses histograms to describe shape features in a log-polar coordinate system that can reflect the distribution of sampling points on the contour well. Stein et al. [20] introduced a 3D splash descriptor by a local reference frame (LRF) to achieve posing stability. Tombari et al. [21] proposed a spherical coordinate system to divide the neighboring points around the query point and obtain the signature of histograms of orientations shot (SHOT) by counting the number in each subspace. This method balances descriptiveness and time efficiency. Rusu et al. [22] proposed a fast point feature histogram (FPFH) of two-point descriptors for all neighboring points of the reference point. It simplifies the point feature histogram (PFH) descriptor and decreases its computational complexity from $O(n^{2})$ to O(n). Flint et al. [23] presented 3D-SIFT descriptor by extending the 2D scale-invariant feature transform (SIFT) descriptor to 3D space. Guo et al. [24] proposed a local feature descriptor of rotation projection statistics (RoPS) by performing a series of operations such as rotation, projection, distribution matrix calculation, statistics analysis, and merging on the neighbor points. Yang J et al. [25] presented a local feature statistic histogram (LFSH) which formed a comprehensive description of the local geometry by statistically encoding the local depth, point density, and angles between normals. Chen et al. [26] introduced a descriptor based on plane/line structural features and achieved good performance of artificial point clouds with regular planes. However, it is not good at handling irregular features such as plants. Based on voxelization, Wang et al. [27] proposed a 3D SigVox descriptor, which is the first shape descriptor of complete objects to match repetitive objects in large point clouds. For more details about 3D point cloud coarse registration, readers are referred to the survey in [24].

### 2.3. Fine Registration

In contrast with the coarse registration, the fine registration method primarily refers to directly obtaining the correspondence between two original points rather than feature descriptors. It produces a more precise result from the initial transformation. The most well-known fine point cloud registration approach is the iterative closest point (ICP) method from Besl and McKay [4]. This method starts from initial alignment and then alternates between establishing correspondences through the closest point and recalculating the alignment according to the current correspondences. The ICP algorithm achieves positive performance in point cloud registration. However, it takes too much time due to the closest point search, and it easily falls into a local minimum, especially when there are noises. Some studies have focused on solving the local minimum problem. For instance, Chetverikov et al. [28] introduced trimmed-ICP, which is based on the consistent use of the least trimmed squares (LTS) in all phases of the operation to improve the robustness to noise. Yang et al. [29] proposed GO-ICP, which is the first variant of ICP to solve the local minimum. However, GO-ICP is still sensitive to occlusion and partial overlap. In addition, Biber et al. [30] proposed the normal distribution transform (NDT), which is applied to the statistical model of three-dimension points rather than a local feature or closest points. Therefore it does not need to include feature calculation and matching of corresponding points in the ICP process, and NDT runs faster. The contributions of some other studies are improving computational efficiency. Chen et al. [31] took advantage of the tendency of most range data to be locally planar and introduced the point-to-plane variant of ICP. Compared with the ICP algorithm, point-to-plane ICP greatly advances operational efficiency. Khoshelham et al. [32] used closed-form point–to–plane correspondence and improved the computing speed. Segal [33] combined the ICP and “point-to-plane ICP” algorithms into a single probabilistic framework called GICP. The factors that affect the computation efficiency are closest point search and the solution of nonlinear problems. Bouaziz et al. [34] used sparsity inducing norms to make the algorithm efficient. Yang J et al. and Koide [35] extended the GICP with voxelization to avoid the costly nearest neighbor search. Due to the popularity of low-cost point cloud acquisition devices such as Kinect and Realsense, RGB-D point cloud processing has become more crucial. Korn M et al. [36] integrated L*a*b color space information into GICP and presented a method to support point cloud registration with color information. The convolutional neural network (CNN) has a strong ability to learn feature descriptors. Aoki et al. [37] expanded the PointNet and LucasKanade (LK) algorithms into a single trainable recurrent deep neural network and achieved outstanding registration performance. However, it is not good at handling partial-to-partial registration. The deep closest point (DCP) proposed by [38] performs well in solving the local minimum problem of ICP, but it can only handle a single object point cloud with less than five thousand points.

## 3. Methodology

An accurate spatial registration method was designed to align the 3D point clouds of different perspectives and partial overlap into the same coordinate system and make the registration error small enough. Figure 1 shows the flowchart of this proposed method. Firstly, the improved voxel filter was used to remove outliers of the original point cloud. Then, the MVGICP was designed to achieve multi-scale iteration for coarse-fine registration end-to-end. Multiple threads accelerated the entire calculation process.

### 3.1. Problem Formulation

Point cloud registration can be described as finding the best affine transformation between two point cloud sets. Given a pair of data *P* and *Q*, for any point $p_{i}\in P$ and $q_{i}\in Q$, the problem can be addressed like this:(1)$$q_{i}=Rp_{i}+T+\epsilon$$
where *R* and *T* are the rotation matrix and transformation vector, respectively. ϵ represents outliers and noises of the raw data. The 3D point cloud registration algorithm should be accurate and fast. It should also be robust to noises, arbitrary poses, and other perturbations. In this section, a registration algorithm that satisfies these qualifications is introduced. It consists of two main parts: point cloud filtering and multi-scale registration.

### 3.2. Pointcloud Filtering

As a benefit from the terrestrial laser scanners (TLS) and low-cost 3D instruments such as Kinect, point clouds can be obtained easily. However, the raw data contain a huge amount of redundant points and noises because of the external interference as well as measurement errors of the collection equipment. Therefore, the original point cloud data need to be effectively filtered and denoised, enhancing the algorithm’s stability to noise and reducing point cloud data. More precisely, this can enhance the accuracy and speed of subsequent point cloud processing. Researchers have proposed a variety of methods to downsample and eliminate outliers in the point cloud. The voxel filter has the highest computational efficiency, but it is weak at filtering outliers. The statistical outlier removal method based on point distribution can achieve good filtering results, but the calculation time is longer. To overcome these shortcomings, this paper proposes an improved voxel filtering method, which strikes a balance between the computational efficiency and the outlier filtering.

As shown in Figure 2, the input point cloud data are divided by a three-dimensional voxel grid. In detail, the distribution range $\left[\left(x_{min},x_{max}\right),\left(y_{min},y_{max}\right),\left(z_{min},z_{max}\right)\right]$ of the point cloud in three dimensions is obtained first of all. Then, an appropriate cell size *c* (Bunny: 0.006; Hippo: 0.02; Chef: 6.) to rasterize the point cloud is selected. The number of grids obtained in *X* dimension can be described as:(2)$$N_{x}=\left(x_{max}-x_{min}\right)/c+1$$

Likewise, the number of grids in $N_{y}$ and $N_{z}$ can be obtained respectively and $\left(N_{x}\times N_{y}\times N_{z}\right)$ cubes can be calculated. If a point cloud *P* containing *N* uniformly distributed points, the number of points per cube is: (3)$$n=N/(N_{x}\times N_{y}\times N_{z})$$

In fact, the density of points in the point cloud is not uniform. As shown in Figure 2, the model has a high point density, and the noise points are low. The model’s cube contains a large number of points, while the noise point cube contains fewer points. In addition, dividing on the three coordinate axes will produce a lot of repeated grids, meaning that the real number of grids will be less than $\left(N_{x}\times N_{y}\times N_{z}\right)$. According to these factors, outliers can be filtered out by setting an appropriate threshold to eliminate cubes with fewer points than the threshold. Moreover, the improved voxel filter is fast and efficient, which greatly improves outlier removal performance only by adding a point density calculation in the traditional voxel filter. The selection of the threshold is the most important step. A too-low threshold will result in incomplete removal of outliers, while a too-high threshold will cause holes in the point cloud. In this paper, the threshold was set as t=2n.

After removing the outliers, *P* is converted to $P^{\prime }$ which contains $N^{\prime }$ points. Then, the centroid of all points in each voxel grid can be calculated:(4)$$c_{i}=\frac{1}{a}\sum_{j=1}^{a}p_{j}\left(i=1,2,...b\right)$$
where *b* represents the number of remaining grids, and *a* is the number of points in each grid. Finally, all the points in each grid are replaced by the obtained centroid point, which can reduce the point cloud scale.

### 3.3. Mvgicp: Point Cloud Registration

In this section, we propose a Multi-Scale Voxelized GICP (MVGICP) method to register point clouds, which is an improved version of GICP and VGICP. Compared with the GICP and VGICP, MVGICP achieves smaller registration errors and faster speeds. The pseudo code for the MVGIGP is given in Algorithm 1, which firstly uses a bigger voxel size (one hundred times the average density of point cloud) to segment the target point cloud and obtain the mean and covariance under the assumption that points within the voxel grid satisfy Gaussian distribution. Then, MVGICP brings the obtained value into the GICP framework to calculate the initial transformation *T*. Afterward, a step-down voxel grid size (the minimum is six times the average density of the point cloud) is adopted to optimize the final result until the threshold or the maximum number of iterations is reached.
**Algorithm 1:** MVGICP
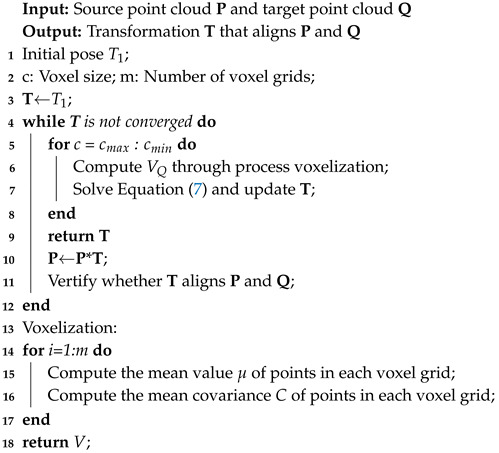


#### 3.3.1. Basic Concept of Gicp

Generalized-ICP (GICP) is a variant of the iterative closest point (ICP) algorithm. It combines the traditional ICP and point-to-plane variant in a probabilistic framework and forms a plane-to-plane matching strategy. As shown in Figure 3, the point-to-point matching of traditional ICP is easily falls into a local minimum. GICP is a distribution-to-distribution strategy. It chooses the covariance matrix of neighboring points instead of each point to calculate the best transformation *T*, which overcomes the point-to-point method’s shortcomings.

On the surface from *P* and *Q*, a point is sampled as a Gaussian distribution: $p_{i}∼N\left(\hat{p_{i}},C_{i}^{P}\right)$, $q_{i}∼N\left(\hat{q_{i}},C_{i}^{Q}\right)$. Due to the similar framework to ICP, the tranformation of GICP can be expressed as:(5)$$T=\arg\max_{T}\sum_{i}\log\left(p\left(d_{i}\right)\right)=\arg\min_{T}\sum_{i}d_{i}^{T}{\left(C_{i}^{Q}+TC_{i}^{P}T^{T}\right)}^{-1}d_{i}$$
where $d_{i}=q_{i}-Tp_{i}$ is the error of alignment point clouds and approaches zero. $C_{i}^{P}$ and $C_{i}^{Q}$ are the covariances of point clouds *P* and $Q^{\prime }$, respectively, which are estimated from their adjacent points. GICP increases calculation speed. However, similar to ICP, it requires a good initial pose and closest point searching, which results in a local minimum and huge calculation costs.

To further optimize GICP, many improvements have been proposed. Three-Dimensional Normal Distribution Transform (3D-NDT) transforms a discrete 3D point set into a piecewise continuous probability density represented by a normal distribution set and maximizes the probability that one distribution falls into another. The use of the probabilistic-based method avoids the closest point searching and improves calculation efficiency. Nevertheless, the accuracy of NDT is relatively low. VGICP adopts the 3D-NDT strategy and extends GICP with voxelization. However, both VGICP and NDT are sensitive to the voxel size. MVGICP overcomes its shortcomings. It can be initially registered at a large voxel scale, and the small size can refine the registration result.

#### 3.3.2. Mvgicp’S Optimization of Local Minimum

Figure 4 shows the concept of MVGICP; its main idea is to voxelize the point cloud with different voxel sizes. In detail, MVGICP firstly subdivides the space occupied by the point cloud model into small cubes of regular size, and then calculates the mean *u* and covariance $C_{i}^{Q}$ of the cube containing multiple points. So, the distance between $p_{i}$ to its adjacent points $q_{i}$ within radius *r*, which can be written as this formula: $d_{i}^{{}^{\prime }}=\sum_{j}\hat{q_{j}}-Tp_{i}$, and Formula (5) can be transferred to:(6)$$T=\arg\min_{T}\sum_{i}{\left(\sum_{j}\hat{q_{j}}-Tp_{i}\right)}^{T}{\left(C_{i}^{Q}+TC_{i}^{P}T^{T}\right)}^{-1}\left(\sum_{j}\hat{q_{j}}-Tp_{i}\right)$$

To improve the computing efficiency, the mean of all neighbor points in the voxel grid is used instead of a single point, and the above objective function becomes:(7)$$T=\arg\min_{T}\sum_{i}N_{i}{\left(\frac{\sum q_{i}}{N_{i}}-Tp_{i}\right)}^{T}{\left(C_{i}^{Q}+TC_{i}^{P}T^{T}\right)}^{-1}N_{i}\left(\frac{\sum q_{i}}{N_{i}}-Tp_{i}\right)$$
where $N_{i}$ is the number of neighbor points in the voxel grid, which is closely related to the selected voxel size. According to the Gauss–Newton method, the objective function composed of $\left(T,p_{i},C_{i}^{P},v.u,v.C,v.N\right)$ can be iteratively updated to get the initial transformation. From Formula (Equation 7), voxelization can smooth the target point cloud’s distribution. Hence, the voxel size is significant. A large size can smooth a larger surface so that the point cloud with larger differences can be aligned. Therefore, MVGICP solves the problem of GICP’s high initial position requirements issues.

#### 3.3.3. Mvgicp’S Fine Registration

For large voxel size, MVGICP can roughly align the point cloud, but its accuracy is low. After obtaining the initial transformation, fine registration is required. As shown in Figure 4b, the number of the points contained in the voxel decrease when reducing the voxel size. This will gradually weaken the target point cloud’s smoothing effect, and save more geometric feature information; thus, higher registration accuracy can be obtained. Therefore, for the small voxel size, MVGICP can register the point cloud more accurately.

In summary, MVGICP does not require a coarse-to-fine registration strategy. Large-scale MVGICP plays the role of initial registration, and gradually reducing the voxel size will further improve the results. Overall, MVGICP runs faster without the closest neighbor search process.

## 4. Experiment and Analysis

In order to verify the accuracy of geometric registration and computational efficiency of the algorithm in this paper, we designed two sets of experiments. The first set of experiments compares the improved voxel filter’s outlier filtering performance and computational efficiency with other filtering algorithms. The second set of experiments compares MVGICP with other methods in terms of the registration performance and time consumption based on different datasets. The registration experiment is conducted based on the open-source PCL. The experiment computer is equipped with Intel i7-6700hq, 2.6 GHz CPU, 16.00 GB of memory, and a 64-bit Ubuntu 18.04 operating system.

### 4.1. Outlier Filtering Result

To validate the filtering algorithm proposed in this paper, the different ratios of random noise are added to the public Bunny, Chew, and Hippo point cloud models. Traditional voxel grid filter (Voxel) [5], StatisticalOutlierRemoval filter (Sor) [6], KNN-based local outlier factor (KLof) and statistical method (KSor) [9], dynamic standard deviation threshold (DSDT) [7] and our method for removing outliers and downsampling are applied, respectively. The KSor and DSDT improve the statistic methods by KNN and dynamic standard deviation threshold, respectively. KLof is mainly based on the local density of the neighbors, while our approach concerns the points within the voxel. The filtering results are shown in Figure 5.

We introduce two evaluation metrics for point cloud denoising. The ground-truth and predicted point cloud are described as $P={\left\{p_{i}\right\}}_{i=1}^{N_{1}}$, $Q={\left\{q_{i}\right\}}_{i=1}^{N_{2}}$, where $p_{i}$, $q_{i}$∈$R^{3}$. $N_{1}$ and $N_{2}$ indicates the number of points and they may not be equal, respectively. The metrics are defined as follows:

(1) Root-mean-square-error (RMSE): RMSE is the square root of mean squared error (MSE). Compared with mean squared error (MSE), RMSE can reduce the magnitude of error between different algorithms and make it easier to characterize the comparison curve. It calculates the Euclidean distance between the ground-truth and predicted point cloud.
(8)$$\begin{array}{cc} MSE & =\frac{1}{2N_{1}}\min\sum_{i=1}^{N_{1}}\sum_{j=1}^{N_{2}}{\left(p_{i}-q_{j}\right)}^{2}+\frac{1}{2N_{2}}\min\sum_{j=1}^{N_{2}}\sum_{i=1}^{N_{1}}{\left(q_{j}-p_{j}\right)}^{2}\\ RMSE & =\sqrt{MSE} \end{array}$$

(2) Signal-to-noise ratio (SNR): SNR is a common indicator to measure the level of image quality. Usually, a higher SNR means better graphics quality. SNR is measured in dB given as:(9)$$SNR=10\log\frac{\frac{1}{N_{2}}\sum_{j=1}^{N_{2}}{\left(q_{j}\right)}^{2}}{MSE}$$

Random points are generated and added to the original models to form point clouds with different noise ratios. Figure 5 shows the outlier removal results of different algorithms. The Sor, KSor and DSDT are unable to remove the outliers. That is because the statistical-based method relies on the standard deviation, which is sensitive to the outliers of the boundary. KLof and our method have a better performance due to the local density calculation. The results demonstrate that the improved voxel filter can remove the different outlier ratios efficiently.

In order to quantitatively evaluate the outlier removal performance of the improved filter, the SNR and the processing time are presented. According to Equation (Equation 9), the original point cloud after voxel filtering is regarded as the ground truth, and SNR of the filtered point cloud is calculated. The compared methods perform outlier removal firstly and then use the same voxel size for downsampling.

Figure 6 shows the SNR and processing time of different filtering methods. The traditional voxel filter performs worst because it only has the ability of downsampling. The Sor, KSor and DSDT are not good at filtering the point cloud with higher outlier ratios. KLof and our method perform well in most cases; this indicates that the density-based method is more suitable in the uniform distribution of outliers. In Figure 6b, due to the KNN process, the processing time of KSor and KLof is significantly higher than that of other methods. Our algorithm achieves superior efficiency by avoiding the closest point calculation. Figure 6 demonstrates that the improved voxel filter balances computational efficiency and outlier filtering.

Another test is proposed to evaluate the effect of filtering on the registration performance quantitatively. The evaluation metric is RMSE. After outlier removal, classical ICP and GICP algorithms are selected to register the two point clouds. Due to the different denoising performance of the algorithm, outliers cannot qualify RMSE to indicate the registration result’s quality. Therefore, in this section, the transformation matrix obtained by denoising the point cloud is applied to act on the ground truth, and then the RMSE is calculated. Table 1 and Table 2 respectively show the RMSE of filtered point cloud after ICP and GICP registration. Corresponding to the filtering results, our method achieves the best performance among the six filtering algorithms. The experimental results show that outlier removal significantly influences the performance of the point cloud registration.

### 4.2. Synthetic Data Registration

The MVGICP registration algorithm is tested on simulated data to prove its robustness to perturbation. ICP [4], GICP [33], VGICP [35], and some other coarse-to-fine strategy methods like Super4PCS-ICP (SICP) [17], Super4PCS-GICP (SGICP) [17,33] are tested as comparison methods. ICP obtains the correspondence between two original points directly. GICP is the first ICP variant algorithm that uses the distribution-to-distribution strategy. VGICP and our method extend GICP with fixed-scale voxels and multi-scale voxels. Coarse-to-fine methods use four-congruent point sets to roughly align the point clouds and further refine the result by ICP or GICP. Simulation data include the Bunny and Dragon from the Stanford dataset and T-rex, Chef, Chicken, and parasaurolophus from the UAW dataset [39].

The stability of MVGICP on Gauss noise is tested first. The scanned point cloud may contain some noise points and lead to multiple errors, such as noise, rotation and translation. The noise points are attached to the point cloud rather than free, so it is not easy to filter them out and affect the registration results. Therefore, the stability of the algorithm to noise perturbation has great significance. As shown in Figure 7, Gauss noise is added to Chef, and the standard deviations are 0, 0.5, 1, and 2, respectively. Since the Gaussian noise is related to the point cloud density, the UAW dataset with a relatively similar point density is selected.

Figure 8 shows the average RMSE (smaller is better) of each method at different noise levels on the UAW dataset. As shown in Figure 8a, ICP and our method are more robust to noise disturbance. SICP and SGICP also achieve good results in the case of low noise interference. Figure 8b–d display similar RMSE trends; ICP, the coarse-to-fine strategy, and our method are relatively stable to noise. Due to the fixed neighborhood points and voxel size, GICP and fixed-scale VGICP are less effective. In summary, MVGICP achieves the best registration effect in all cases, demonstrating its good robustness to the noise perturbation. ICP and the coarse-to-fine method obtain RMSE with similar values. Moreover, by comparing Figure 8a and the other three figures, the registration results of these algorithms on different models are not the same. Different model structures or surface features may cause this, but our method achieves the best results in all structural models

To further evaluate the algorithms, a comparison of the above algorithms’ processing time is also performed. As shown in Table 3, due to the influence of the initial registration, the coarse-to-fine method takes the most time. In addition, since voxelization can smooth the point cloud’s surface, VGICP is more effective than ICP and GICP. With multi-scale voxel division and multi-level iterations, our method can quickly obtain a good initial transformation through the large-sized voxel. This appropriate initial transformation can save the convergence time for a small-sized voxel. Based on this, our algorithm has the highest computational efficiency. It can be concluded that MVGICP works well on the point clouds with different noise ratios.

The algorithms’ performance for registering varying degrees of rotations between a pair of point clouds is also tested. To perform a controlled evaluation, the degree of rotation between any two point clouds is given. The viewing angle differences of the input clouds are from $15^{\circ }$ to $90^{\circ }$, and its step size is $15^{\circ }$. Figure 9 shows the results of the RMSEs of several algorithms on different rotation perturbations. The RMSE of GICP increases significantly as the pose difference becomes larger. ICP is more stable than fixed-scale VGICP. That is because the VGICP relies on a certain voxel size, which is susceptible to sub-voxel misalignment. Both our algorithm and the coarse-to-fine method are relatively stable to rotation disturbances. Nevertheless, compared with the coarse-to-fine strategy, our algorithm is more concise and effective because our algorithm does not require initial registration. This confirms that MVGICP can register the point clouds with various rotations, which is important in the 3D reconstruction.

Figure 10 shows the visual registration results of the Bunny and the Dragon models with different rotations. The Bunny and the Dragon rotates on the z-axis and x-axis, respectively, where the rotation of Bunny is $30^{\circ }$, and the rotation of Dragon is $15^{\circ }$. It can be observed that MVGICP finely registers point clouds with different rotations.

### 4.3. Multi-View Registration

In this section, the multi-view registration results of our algorithm on the synthetic data and real data are presented.

#### 4.3.1. Multi-View Synthetic Data Registration

(1) Multi-view registration without outliers: the incremental registration ability of our algorithm on the dataset UAW is evaluated. Multiple scanned point clouds of the Chicken and T-rex without outliers are used as input. Figure 11 shows the multi-view point clouds and the registration results. As shown in Figure 11a,c, each model has 12 scans, and each scan in the point clouds to be registered is remarked by a unique color. The benefits from multi-scale iteration, although these point clouds have different rotations and overlap ratios, include that they are all well aligned.

In order to quantitatively analyze the multi-view registration capability of our algorithm, the average registration errors of each object are presented in the Table 4, and are defined as the average difference from the ground truth. ICP is selected for comparison in this part. The table shows that the rotation and translation errors of MVGICP are less than ICP. Since ICP relies on the location of every point, it is easy to fall into local optima. Furthermore, the experimental results demonstrate that MVGICP can accurately register point clouds from multiple views.

(2) Multi-view registration with outliers: To verify the whole algorithm proposed in this article, the Armadillo, with outliers, will be tested. Figure 12 shows the input outlier point clouds and their registration results. As shown in Figure 12a, the scans of the Armadillo from different angles are placed together with many outliers. Figure 12b–d shows the registration results in three different views respectively. The registration results indicate that the proposed algorithm filters out the outliers with high accuracy.

To evaluate the influence of outlier filtering on the final result, we present the average registration errors of Filtered-MVGICP and MVGICP in Table 5. The rotation and translation errors of MVGICP are nearly 10 times higher than filtered MVGICP. This strongly supports the idea that the proposed method greatly reduces the registration error and shows the stability for outlier 3D point cloud registration.

#### 4.3.2. Multi-View Real Data Registration

Figure 13 shows the registration results of MVGICP on the real outdoor and indoor scenes. Compared with the model data, the scene point cloud is larger and has more geometric features. There are obvious pose differences between the point clouds to be registered. Furthermore, the results indicate that accurate registration for real scene data is accomplished, and scans are well converged by MVGICP.

## 5. Conclusions

This paper presents an efficient, stable, and accurate hierarchical optimization algorithm for 3D point cloud registration. Our improved voxel filter can remove outliers well but also has good computational efficiency. In addition, MVGICP is effective at finding optimal transformation at a different level of noise and rotation perturbation and effectively handling various types of point cloud, such as model, scene scans, TLS, and RGB-D, etc. The implementation of a multi-threaded operation can achieve accelerated calculation without affecting the quality of the final result. Our algorithm solves the problems raised in the introduction extremely well. It may be broadly applicable in 3D reconstruction, computer vision, and robotics.

The proposed method still has some limitations. The algorithm in this paper can be further improved for registering large translation disturbances and low overlap scene data. We will consider adding local features information into the framework to improve handling point clouds in complex scenes.

## Figures and Tables

**Figure 1 sensors-20-06999-f001:**
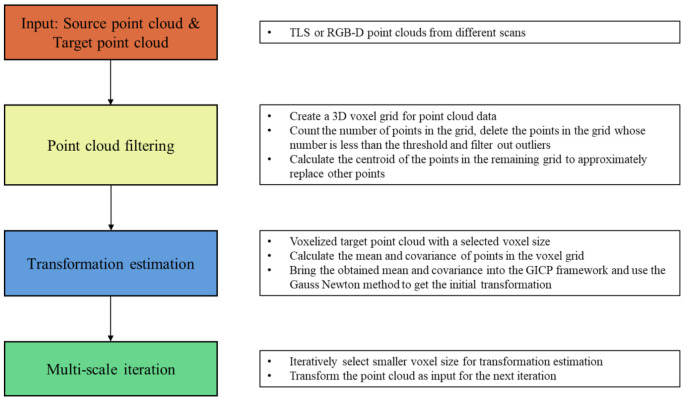
Flowchart of the proposed registration algorithm for a 3D point cloud.

**Figure 2 sensors-20-06999-f002:**
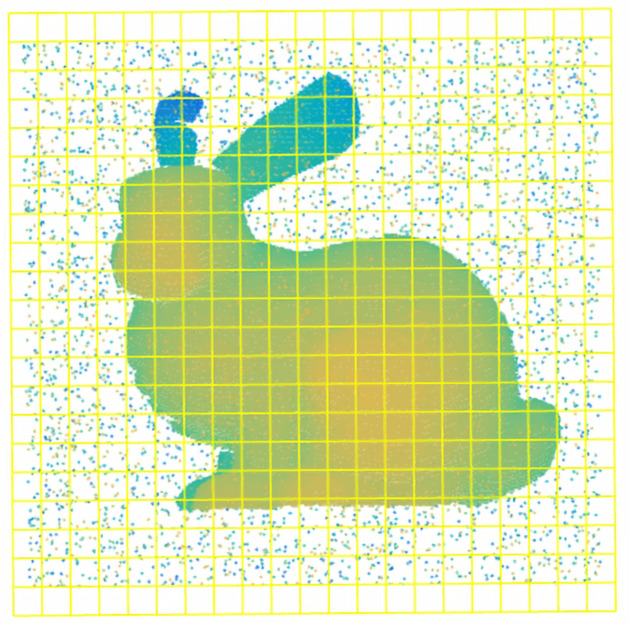
Illustration of the improved voxel filter. The density of points in different voxel grids is not equal, so the threshold can be set to eliminate outliers.

**Figure 3 sensors-20-06999-f003:**
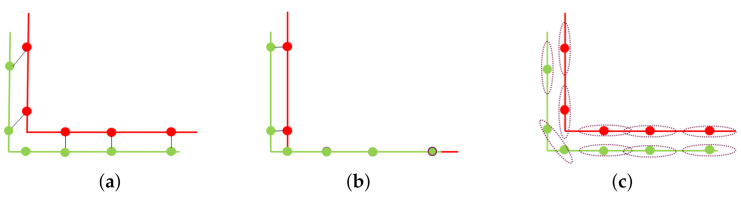
Illustration of point-to-point and plane-to-plane strategies. (**a**) Point-to-point strategy; (**b**) point-to-point is easily trapped into local optima; (**c**) plane-to-plane strategy.

**Figure 4 sensors-20-06999-f004:**
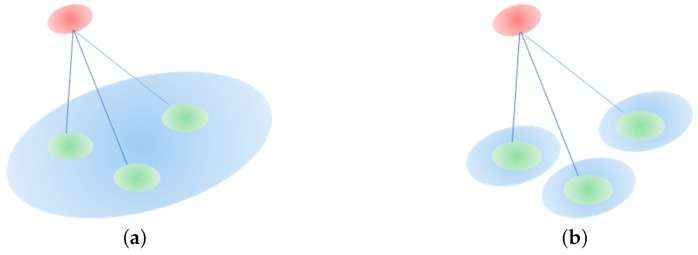
Illustration of MVGICP. (**a**,**b**) represent the corresponding models of large- and small-scale MVGICP distance calculation, respectively. The red, green, and blue circles represent the source point cloud distribution, target point cloud distribution, and voxel, respectively.

**Figure 5 sensors-20-06999-f005:**
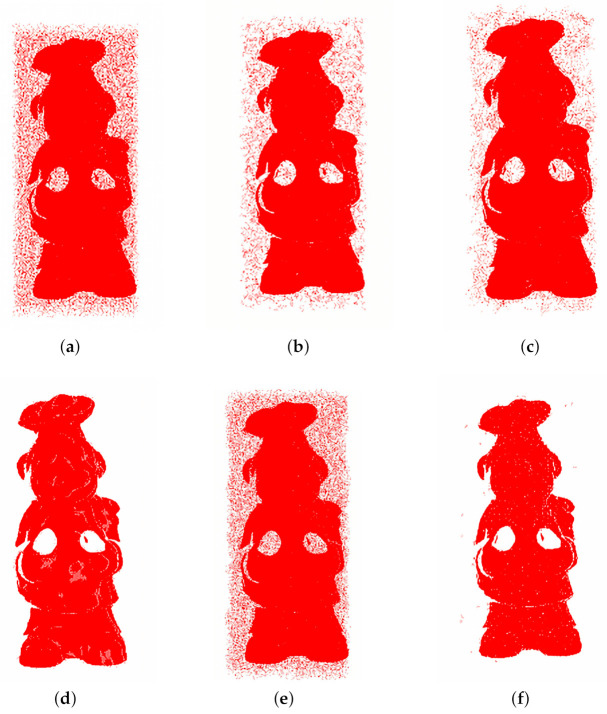
The outlier removal results of different algorithms. (**a**) Chef with 0.5 ratio of random outliers; (**b**) filtering result with Sor; (**c**) KSor; (**d**) KLof; (**e**) DSDT; (**f**) our method.

**Figure 6 sensors-20-06999-f006:**
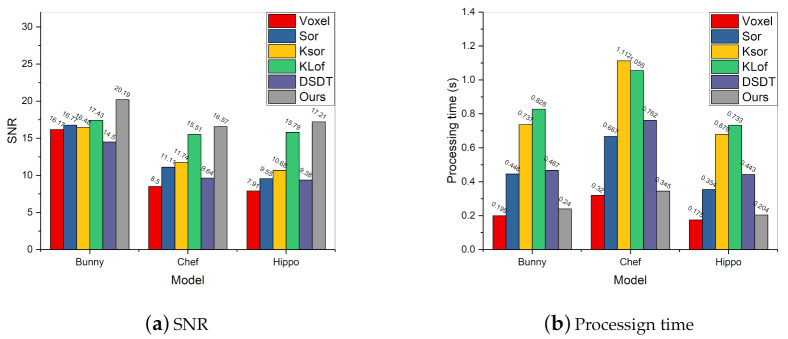
Evaluation of signal-to-noise ratio (SNR) and processing time on several filtering algorithms. Our approach obtains the highest SNR, and its computational efficiency is much higher than other methods.

**Figure 7 sensors-20-06999-f007:**
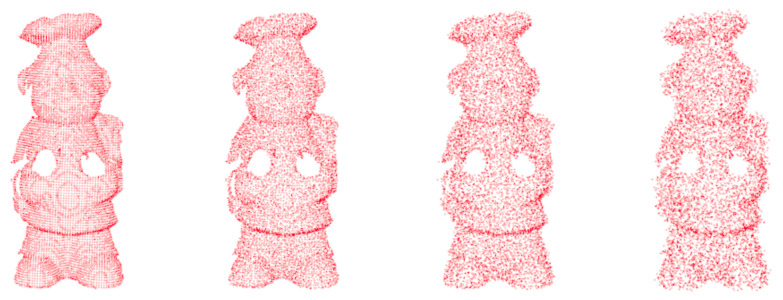
Different Gaussian noise levels are on Chef, the standard deviations from left to right are 0, 0.5, 1, 2, respectively.

**Figure 8 sensors-20-06999-f008:**
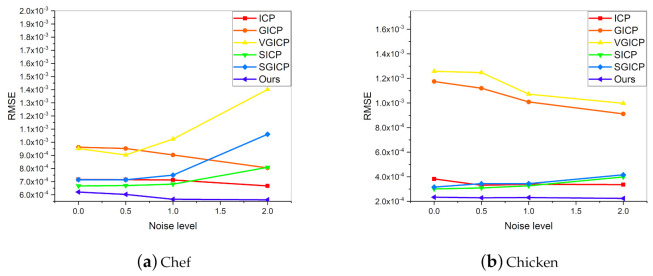
Average RMSE of several methods at different noise levels on the UAW dataset.

**Figure 9 sensors-20-06999-f009:**
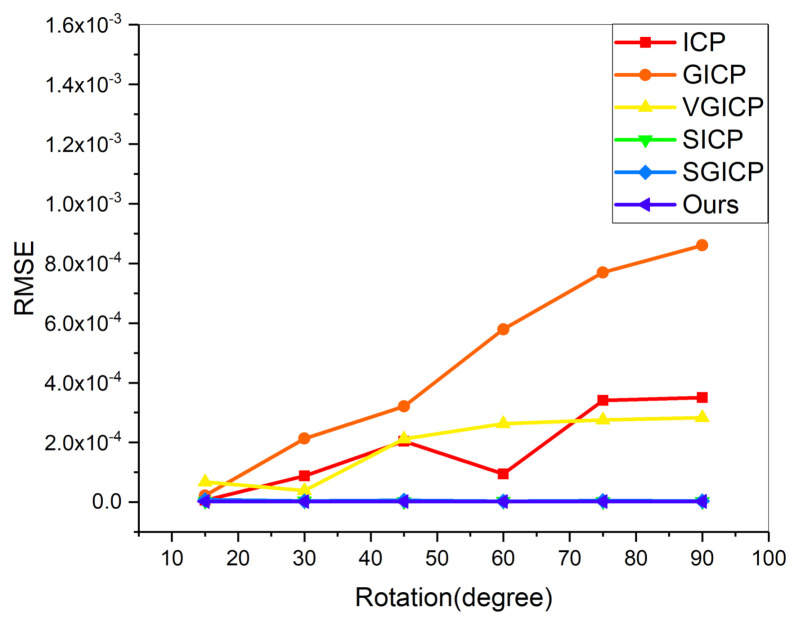
Controlled comparison between algorithms on rotation perturbation. ICP, Super4PCS-ICP (SICP), Super4PCS-GICP (SGICP) and our method obtain almost the same RMSE.

**Figure 10 sensors-20-06999-f010:**
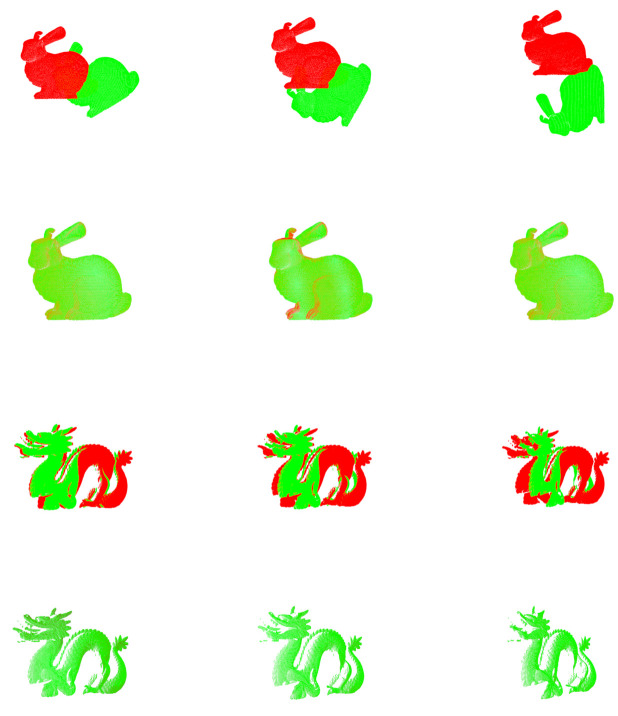
Visual registration results of the Bunny and Dragon models with different rotations.

**Figure 11 sensors-20-06999-f011:**
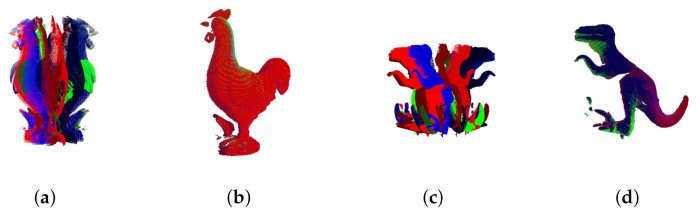
Multi-view registration through our approach on chicken and T-rex model of the UWA dataset. (**a**,**c**) are input scans from different views and (**b**,**d**) are the registration results.

**Figure 12 sensors-20-06999-f012:**
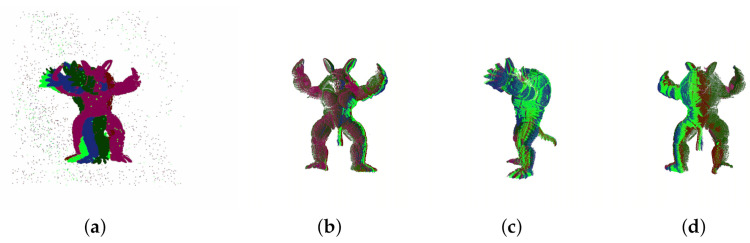
Multi-view registration through our approach to the outlier Armadillo model of Stanford dataset. (**a**) Multi view Armadillo point clouds with outliers; (**b**–**d**) are the registration results in three views.

**Figure 13 sensors-20-06999-f013:**
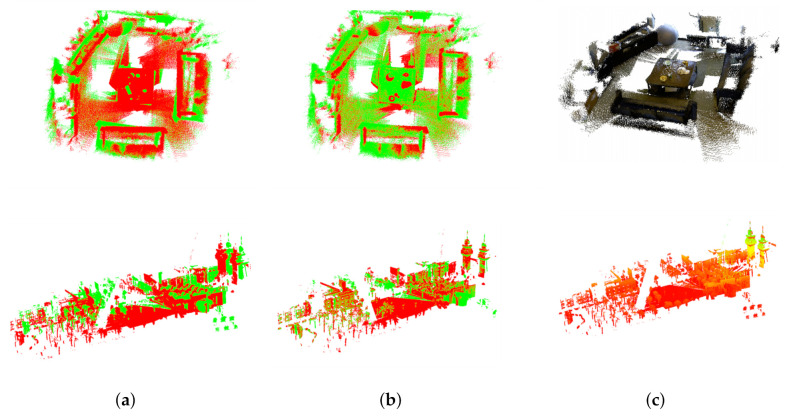
Multi-view registration through our approach to scene data. (**a**) Input point clouds; (**b**) the registration results; (**c**) RGB rendering results.

**Table 1 sensors-20-06999-t001:** Root mean square error (RMSE) comparison of generalized iterative closest point (GICP).

	Groundtruth	Voxel	Sor	KSor	KLof	DSDT	Ours
Bunny	$3.97\times 10^{-3}$	$2.69\times 10^{-2}$	$8.43\times 10^{-3}$	$7.52\times 10^{-3}$	$5.53\times 10^{-3}$	$6.77\times 10^{-3}$	$3.56\times 10^{-3}$
Chef	$3.92\times 10^{-1}$	$4.41\times 10^{-1}$	$4.32\times 10^{-1}$	$4.23\times 10^{-1}$	$3.93\times 10^{-1}$	$4.33\times 10^{-1}$	$3.93\times 10^{-1}$
Hippo	$3.02\times 10^{-2}$	$6.03\times 10^{-2}$	$5.94\times 10^{-2}$	$6.54\times 10^{-2}$	$4.77\times 10^{-2}$	$6.18\times 10^{-2}$	$3.03\times 10^{-2}$

**Table 2 sensors-20-06999-t002:** RMSE comparison of GICP.

	Groundtruth	Voxel	Sor	KSor	KLof	DSDT	Ours
Bunny	$6.50\times 10^{-3}$	$2.70\times 10^{-2}$	$2.21\times 10^{-2}$	$2.55\times 10^{-2}$	$8.75\times 10^{-3}$	$2.59\times 10^{-2}$	$6.24\times 10^{-3}$
Chef	$4.34\times 10^{-1}$	$4.41\times 10^{-1}$	$4.35\times 10^{-1}$	$4.53\times 10^{-1}$	$4.36\times 10^{-1}$	$4.38\times 10^{-1}$	$4.35\times 10^{-1}$
Hippo	$3.13\times 10^{-2}$	$6.04\times 10^{-2}$	$5.78\times 10^{-2}$	$5.84\times 10^{-2}$	$4.53\times 10^{-2}$	$5.97\times 10^{-2}$	$3.14\times 10^{-2}$

**Table 3 sensors-20-06999-t003:** Average processing time (s) of several methods at different noise levels on UAW dataset.

	ICP	GICP	VGICP	SICP	SGICP	Ours
Chef	12.22	10.98	8.64	14.33	11.20	2.39
Chicken	15.74	10.95	10.27	14.66	15.69	2.77
Parasaurolophus	13.81	9.84	8.51	14.39	17.71	2.28
T-rex	10.53	10.88	7.61	11.74	17.54	2.00

**Table 4 sensors-20-06999-t004:** Multi-view registration results on the Chicken and T-rex models of the UWA dataset.

Model	Chicken		T-Rex	
Method	MVGICP	ICP	MVGICP	ICP
Error $\epsilon_{r}\left(\circ \right)$	0.1952	0.238	0.0563	0.0712
Error $\epsilon_{t}\left(d_{res}\right)$	0.1491	0.250312	0.0707	0.0871

**Table 5 sensors-20-06999-t005:** Registration results of Filtered-MVGIP and MVGICP to the outlier Armadillo model.

Method	Filtered-MVGICP	MVGICP
Error $\epsilon_{r}\left(\circ \right)$	0.9267	9.278
Error $\epsilon_{t}\left(d_{res}\right)$	$8.39\times 10^{-4}$	0.0085

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
