# Peer review of "Hierarchical Optimization of 3D Point Cloud Registration"

_sensors, 2020, doi:10.3390/s20236999_

Round 1

Reviewer 1 Report

In this work, the authors have presented proposed a hierarchical optimization approach that included improved voxel filter and Multi-scale Voxelized Generalized-ICP (MVGICP) for 3D point cloud registration. Combining traditional voxel sampling with point density, the proposed achieve the ability to filter outliers and downsample. Due to multi-scale iteration and avoiding closest point computation, MVGICP solves the local minimum problem and increases operating efficiency. In general, I would say this manuscript makes an adequate contribution. However, I have some essential remarks and concerns for the current manuscript, so I cannot recommend accepting this manuscript before flaws are addressed.

1. My first concern is about the contribution of your work. You claimed that you have improved the classic ICP method with a novel filtering approach. However, there are no clear statements in the manuscript about what contribution you have done. So, please highlight your contribution as an independent paragraph.

2. From the title, I feel that voxelization plays a vital role in your method, but after I checked the method, I found that voxelization serves only filtering. So, what is the motivation of conducting such a processing step? Why not just simply conduct conventional downsampling? Moreover, how to select an adequate resolution for voxelization?

3. When you using voxelized point clouds as basic elements for registration, how to handle the sub-voxel accuracy alignment?

4. In your manuscript, what is the difference between Go-ICP and GICP? Are they the same? Please clarify. If so, there is no need to introduce Go-ICP again in Section 3.3.1.

5. In the review part, the entire literature review seems comprehensive, but the mentioned methods are not categorized and structurally discussed. You classified registration methods into fine or coarse registration, but what does fine registration mean? Which level of accuracy can be regarded as fine registration?

6. Some newest methods are missing, especially those deep learning-based ones. Moreover, some recently published papers using voxelization and structural elements for point cloud registration are missing. Please refer to and discuss the following papers in your review part:

Khoshelham, K. (2016). Closed-form solutions for estimating a rigid motion from plane correspondences extracted from point clouds. ISPRS Journal of Photogrammetry and Remote Sensing114, 78-91.

Wang et al. (2017). SigVox–A 3D feature matching algorithm for automatic street object recognition in mobile laser scanning point clouds. ISPRS Journal of Photogrammetry and Remote Sensing128, 111-129.

Xu et al. (2019). Pairwise coarse registration of point clouds in urban scenes using voxel-based 4-planes congruent sets. ISPRS journal of photogrammetry and remote sensing151, 106-123.

Chen et al. (2019). PLADE: A Plane-Based Descriptor for Point Cloud Registration With Small Overlap. IEEE Transactions on Geoscience and Remote Sensing58(4), 2530-2540.

Aoki et al. (2019). Pointnetlk: Robust & efficient point cloud registration using pointnet. In Proceedings of the IEEE Conference on Computer Vision and Pattern Recognition (pp. 7163-7172).

Huang et al. (2020). Temporal comparison of construction sites using photogrammetric point cloud sequences and robust phase correlation. Automation in Construction117, 103247.

7. The experimental evaluation and discussions are not satisfying. A practical registration method must be applicable for large amounts dataset, however, the experiments are conducted on point clouds of small single objects. This is not convincing. So please add additional experiments using large scale datasets, for example, RESSO and WHU-TLS datasets:

RESSO : https://3d.bk.tudelft.nl/liangliang/publications/2019/plade/resso.html

WHU-TLS :  https://www.isprs.org/news/announcements/details.aspx?ID=214

These two datasets are featured with investigations on the overlap ratios and multiple-scans, respectively. I would like to suggest the authors referring these two datasets as a baseline in your current manuscript. Otherwise, you cannot say that your proposed method is efficient for large scale scenarios.

8. Analysis and discussion of experimental results are kinds of too data. There should be some analysis of errors and misalignment, not just a comparison of results. Moreover, from the visualized results, it seems the proposed method reveals a promising performance. However, from the quantized results in tables, it seems there is no significant improvement. What is the reason? Please give a discussion on the achieved results and draw a corresponding conclusion from the facts.

9. For the design of your experiments, I would say that comparing with noise level, overlap ratios are a more critical factor that influences the registration results. However, it seems that you didn't consider this point in experiments. This is also the reason why I recommend you test RESSO or WHU-TLS dataset.

10. Figures in the main manuscript are too small, and some of the words and terms cannot even be recognized. Please recreate these figures for a better resolution and larger size.

11. English writing of this manuscript must be improved. There are plenty of spelling grammar errors. There are also some vague sentences that are difficult to read and understand. A thorough check and improvement of English writing by native speakers would be helpful.

Reviewer 2 Report

This paper proposes a new method for outlier filtering as an improvement of voxel filtering and 3D point cloud registration based on Multi-scale Voxelized Generalized-ICP.

I find this paper well written, net of some misprints and minor spell errors.
I suggest to avoid the use of personal sentences. 

I suggest to better describe the new voxel filter from row 155 to row 163. In particular explain how the new filter strikes a balance between computational efficiency and outlier filtering. 

In figure 3 please put a label on each part of the figure (a,b,c) and explain each part in the caption.

In table 3 please specify the units.

Round 2

Reviewer 1 Report

All my concerns have been addressed. Now I would like to recommend acceptance for this manuscript.

Author Response

Thank you very much for your approval and have a nice day!